# Potential Beneficial Effects of Naringin and Naringenin on Long COVID—A Review of the Literature

**DOI:** 10.3390/microorganisms12020332

**Published:** 2024-02-04

**Authors:** Siqi Liu, Mengli Zhong, Hao Wu, Weiwei Su, Yonggang Wang, Peibo Li

**Affiliations:** 1Guangdong Engineering and Technology Research Center for Quality and Efficacy Re-Evaluation of Post-Market Traditional Chinese Medicine, State Key Laboratory of Biocontrol, Guangdong Provincial Key Laboratory of Plant Resources, School of Life Sciences, Sun Yat-sen University, Guangzhou 510275, China; liusq67@mail2.sysu.edu.cn (S.L.); zhongmli@mail2.sysu.edu.cn (M.Z.); wuhao_cpu@126.com (H.W.); lsssww@mail.sysu.edu.cn (W.S.); wangyg@mail.sysu.edu.cn (Y.W.); 2Maoming Branch, Guangdong Laboratory for Lingnan Modern Agriculture, Maoming 525000, China

**Keywords:** long COVID, SARS-CoV-2, COVID-19, naringin, naringenin, beneficial effects

## Abstract

Coronavirus disease 2019 (COVID-19) caused a severe epidemic due to severe acute respiratory syndrome coronavirus-2 (SARS-CoV-2). Recent studies have found that patients do not completely recover from acute infections, but instead, suffer from a variety of post-acute sequelae of SARS-CoV-2 infection, known as long COVID. The effects of long COVID can be far-reaching, with a duration of up to six months and a range of symptoms such as cognitive dysfunction, immune dysregulation, microbiota dysbiosis, myalgic encephalomyelitis/chronic fatigue syndrome, myocarditis, pulmonary fibrosis, cough, diabetes, pain, reproductive dysfunction, and thrombus formation. However, recent studies have shown that naringenin and naringin have palliative effects on various COVID-19 sequelae. Flavonoids such as naringin and naringenin, commonly found in fruits and vegetables, have various positive effects, including reducing inflammation, preventing viral infections, and providing antioxidants. This article discusses the molecular mechanisms and clinical effects of naringin and naringenin on treating the above diseases. It proposes them as potential drugs for the treatment of long COVID, and it can be inferred that naringin and naringenin exhibit potential as extended long COVID medications, in the future likely serving as nutraceuticals or clinical supplements for the comprehensive alleviation of the various manifestations of COVID-19 complications.

## 1. Introduction

COVID-19 encompasses a range of illnesses resulting from the contraction of severe acute respiratory syndrome coronavirus 2 (SARS-CoV-2). It is spread in the population mainly through droplets, but also via aerosol transmission [1], contamination transmission [2], and the fecal–oral route of transmission [3]. SARS-CoV-2 binds to the angiotensin-converting enzyme II (ACE2) receptor in the body via the S protein, which then invades human tissues and releases RNA via cytotoxicity [4]. The ACE2 receptor is mainly found in type II alveolar cells and is also found in many tissues, such as the heart, esophagus, kidneys, and bladder [5]. Therefore, SARS-CoV-2 invades the human body and causes a multi-tissue disease, including respiratory symptoms. SARS-CoV-2 has a high rate of transmission. The majority of patients experience acute symptoms within a span of 2–14 days following viral exposure. The symptoms are primarily fever, cough, sore throat, myalgia, and an altered sense of smell/taste [6,7]. In addition to cardiac disease [8], neurologic disease [9] and gastrointestinal disease are also seen in critical patients [10]. Most acute symptoms subside within 2–3 weeks of being infected with the virus, although subsequent research has revealed that certain individuals do not completely recuperate from their ailment, but instead, experience post-acute consequences of SARS-CoV-2 infection (PASC), commonly referred to as long COVID [11].

Long COVID is characterized by the presence of persistent, relapsing, new symptoms, or other health effects that manifest following acute SARS-CoV-2 infection [12]. Approximately half of the infected individuals worldwide will develop long COVID, especially in Asia (51%), Europe (44%), and the United States of America (31%), and this figure could be higher due to insufficient statistics and a lack of standardized scoring [13,14]. The symptoms of long COVID are more widespread and have a greater impact on human health and working ability [15]. Statistical analysis of the symptoms of COVID-19 patients one year after infection found that the main symptoms of long COVID include dizziness, fatigue, sleep difficulties, memory loss, anxiety and depression, hair loss, smell and taste disorder, decreased appetite, thirst, chronic cough, pulmonary fibrosis, palpitations, chest pain, diarrhea or vomiting, gastrointestinal disorders, diabetes, changes in sexual desire or capacity, joint and muscle pain, post-exertional malaise, abnormal movements, and thrombus formation [12,16] (Figure 1). Among them, fatigue, memory loss, multiorgan abnormalities, pulmonary fibrosis, heart damage, sleep difficulties, and anxiety and depression will continue unimproved for four months up to two years [17,18]. Furthermore, women and individuals with chronic underlying conditions are more susceptible to the detrimental effects of long COVID, and the intensity of symptoms can be affected by vaccination and the mutated strains of the virus [19]. Current research suggests that the main pathogenic mechanisms of long COVID are the persistence of the SARS-CoV-2 virus [20], microthrombosis [21], immune dysregulation [22], the disruption of mitochondrial function [23], the activation of pathogens such as Epstein–Barr virus (EBV) and human herpesvirus 6 (HHV6) [24], changes in the body microbiota [25], persistent inflammation [26], the activation of autoimmunity [27], brainstem signal transduction, and vagal signaling dysfunction [28,29].

At present, the primary treatment for long COVID is based on distinct symptoms, and there are no definitive medications and therapies for long COVID treatment [30]. However, a large number of studies have identified several potential therapeutic agents against long COVID in the natural constituents of medicinal plants, including the organic sulfur compound from garlic (*Allium sativum* L.) [31], curcumin from *Curcuma longa* [32], nigellone and α-hederin from *Nigella sativ* [33,34], phytocompound 6-gingerol from *Zingiber officinale* [35], and flavonoids from garlic, propolis, and honey [31,36]. In particular, natural and semi-synthetic flavonoids and neutrophil elastase inhibitors isolated from natural sources show great advantages in fighting SARS-CoV-2 or treating long COVID [37,38,39].

Naringin and naringenin are two of the most common flavanones found in citrus plants (Figure 2). They are known to cause bitterness in citrus fruits and have a wide range of biological functions. Naringin is the glycoside form of naringenin. After the oral ingestion of naringin or naringenin by the human body, naringin is converted to naringenin in the presence of intestinal microorganisms and absorbed in the intestinal epithelium [40]. The efficacy of naringin and naringenin is due to anti-inflammatory [41], anticancer [42], antioxidant properties [43,44], treatment of cardiovascular disease [45], antiviral [36] and immune-modulation effects [41], which may play an effective role in treating long COVID. The anti-inflammatory and antioxidant effects of naringenin can effectively counteract the cytokine storm induced by SARS-CoV-2, especially targeting and inhibiting interleukin(IL)-6, the main pro-inflammatory factor of COVID-19 [46]. Moreover, numerous molecular docking studies reveal that naringenin binds to viral spiking proteins, viral major proteases, host receptors, and host viral transport channels and has multiple antiviral effects against SARS-CoV-2 [47]. For instance, naringenin can influence the replication of the viral genome by attaching to SARS-CoV-2 macrodomain RNA polymerase (NSP3), SARS-CoV-2 RNA-dependent RNA polymerase (NSP12), and 3-chymotrypsin-like protease, as well as affect viral invasion by attaching to the ACE2 receptor to achieve an antiviral effect [48,49]. However, despite the abundance of evidence suggesting that naringin and naringenin can be more effective in treating various illnesses, there is yet to be a review of their combined potential in treating multiple coexisting conditions of long COVID. The purpose of this research was to assess the potential beneficial effects of naringin and naringenin in treating long COVID.

## 2. Therapeutic Potential of Naringin and Naringenin in Long COVID

In the following, we will discuss the pathogenesis of 11 common symptoms in patients with long COVID and make reasonable assumptions about the potential therapeutic role of naringin and naringenin in treating these symptoms (Figure 3).

### 2.1. Cognitive Dysfunction

Cognitive dysfunction is an extremely common persistent psychiatric manifestation after COVID-19 referred to as “brain fog”. It is characterized by attention/processing speed deficits, mainly in memory and executive functioning [50]. Fatigue, sleep disturbances, language disorders, and loss of smell/taste accompany it [51,52]. According to meta-statistics, cognitive dysfunction was found to be present in 22% of COVID-19 patients 19 weeks after diagnosis, and the symptom was continued in 19.7% at 9 months after infection [53,54]. The condition is seen in both mildly and severely ill patients and persists for long periods without relief, and the longer the duration of symptoms, the greater the cognitive impairment [55]. The main way that COVID-19 survivors experience cognitive decline is through the extended inflammation caused by SARS-CoV-2, which can increase blood–brain barrier permeability and activate microglia and astrocyte subtypes, leading to cellular stress and neuronal damage [56,57]. The invasion of SARS-CoV-2 also causes midbrain dopamine neuron senescence, which has been implicated in Parkinson’s disease [58]. The downregulation of new neuron formation caused by long-term inflammation leads to reduced synaptic plasticity in the hippocampus, resulting in memory loss in COVID-19 survivors [59]. Moreover, some of the pro-inflammatory factors are involved in mood regulation, leading to depression and other mood abnormalities in survivors [60]. In addition, SARS-CoV-2 directly infects the nervous system and causes neurological destruction mediated by neurological inflammation [61,62]. The virus directly damages the olfactory epithelium via neuropilin-1 and ACE2, thereby affecting the olfactory neural network and causing olfactory malfunction [63]. Patients have exhibited notable enhancements in IL-6, CD70, C-reactive protein (CRP), C-C Motif Chemokine Ligand 11 (CCL11), serotonin, and serum biomarkers of neuronal and gliotic degeneration in the blood, as well as alterations in the brain’s microstructure and functional brain integrity, as indicated by neuropsychological tests, blood tests, and diagnostic brain imaging [64,65,66].

Naringin and naringenin have been shown to provide protection and potentially offer therapeutic benefits against cognitive decline in neurological disorders like Alzheimer’s disease [67,68]. Therefore, they can be reasonably hypothesized to have an ameliorative effect on cognitive dysfunction after COVID-19. Naringin shows a protective effect on the nervous system, mainly through the modulation of glial cell activation and protection against nervous system stresses. Microglia and astrocytes are both essential for safeguarding neurons. Microglia have two activation phenotypes: the M1 type is pro-inflammatory and damages neurons, while the M2 phenotype exerts anti-inflammatory and neuronal-repair functions. Naringin can promote microglia activation toward the M2 type and inhibit its activation toward the M1 type by modulating the janus kinase/signal transd ucer and activator of transcription 3 (JAK/STAT3) signaling pathway [69]. Astrocytes are protective of neuronal cells [70]. Naringin regulates the expression of nuclear factor erythroid 2-related factor 2 (Nrf2), thereby enhancing neuronal protection by astrocytes [71]. Naringenin combination therapy attenuates the pro-inflammatory activation of astrocytes and significantly attenuates 3-nitro propionic acid-induced neuronal cell death [72]. Naringin and naringenin have been shown to protect the nervous system from oxidative stress and inflammation in various studies. Naringenin reduces the production of reactive oxygen species, modulates pro-inflammatory cytokines (IL-1β, IL-6, and (tumor necrosis factor)TNF-α) and anti-inflammatory cytokines (IL-10 and IL-4) to reduce inflammation, and enhances synaptic plasticity by increasing the expression of N-methyl-D-aspartate receptors associated with learning and memory [73]. Naringin significantly attenuates D-galactose, doxorubicin, and Bisphenol A-induced oxidative stress in the nervous system and improves cognitive performance after treatment [74,75,76]. Naringenin exerts a protective effect against lead damage to the nervous system by maintaining the antioxidant enzyme system (superoxide dismutase (SOD), catalase (CAT), and glutathione (GSH)) and inhibiting the elevation of inflammatory factors (NF-κB) and proapoptotic-related protein (Bcl-2 and caspase-3) [77].

Moreover, naringenin has depression-relieving effects. Depression disorders are linked to decreased levels of brain-derived neurotrophic factor (BDNF), which is controlled by the cyclic adenosine monophosphate (cAMP)-cAMP response element binding protein (CREB)-BDNF signaling pathway [78,79]. Naringenin reverses the reduction in BDNF expression induced by high cortisol levels and modulates the mitochondrial apoptotic pathway to inhibit hippocampal apoptosis [80]. Naringenin can potentially enhance the endocrine nervous system by controlling the levels of glucocorticoid receptors and monoamines in the hippocampus, resulting in an antidepressant impact [81]. Furthermore, naringenin and SARS-CoV-2 can cross the blood–brain barrier and enter brain tissue [73]. The combination of naringenin and naringin, which block primary viral proteases, decrease receptor activity, and attach to viral spiny proteins, could potentially lessen neurological harm [48,82].

### 2.2. Immune Dysregulation

SARS-CoV-2 invasion activates the body’s innate and adaptive immunity to clear the invading pathogen [83]. Patients with long COVID have exhibited changes in immune cells, increased levels of autoantibodies, and the reactivation of dormant viruses [27,84,85]. The long-term effects of COVID-19 on the patient’s immune system can be seen through alterations in the epigenetic and transcriptional makeup of monocytes, which can result in long-term inflammation, excessive autoimmune activity, and long-term consequences in the body [86]. Intermediate monocyte abnormalities and T-cell activation were found in recovering COVID-19 patients, which were associated with persistent endothelial cell activation and coagulation dysfunction [87,88]. Patients with long COVID exhibited a decline in cluster of differentiation (CD)4^+^ and CD8^+^ effector memory cells [22], and the expression of the pro-inflammatory cytokines interferon (IFN)-β and IFN-λ1 remained high during the 8 months after infection [89]. The persistence of SARS-CoV-2 also leads to a lack of dendritic cells in the host [90], along with a reduction in non-classical monocyte and lymphocyte subsets [91]. The persistent aberrant activation of immunity produces a broad spectrum of self-targeting antibodies and is hypothesized to be associated with different long COVID symptoms [92]. Long COVID patients exhibit abnormally high levels of functional autoantibodies targeting different G protein-coupled receptors, which are associated with persistent neurological and cardiovascular symptoms in patients [93]. SARS-CoV-2-induced autoimmunity causes immune blood disorders, antiphospholipid syndrome, systemic lupus erythematosus, vasculitis, acute arthritis, and Kawasaki-like syndromes [94].

Naringenin has a palliative effect on T cell-mediated autoimmune diseases [95,96]. Following antigen induction, CD4^+^ T cells differentiate into distinct subpopulations, encompassing pro-inflammatory T cells and anti-inflammatory regulatory T cells, which maintain equilibrium within healthy organisms [97]. Naringenin modulates the immune system’s response to autoantigens by influencing the growth and specialization of T cells, as well as cytokine signaling [98,99]. Naringenin effectively alleviates symptoms of rheumatoid arthritis in rats by modulating lymphocyte polarization, primarily through the reduction in T helper (Th)1 and Th17 cell differentiation and the reduction in IL-6 and TNF-α levels [99,100]. Similarly, naringenin alleviates symptoms of multiple sclerosis and systemic lupus erythematosus by inhibiting the growth and specialization of harmful pro-inflammatory T cells while maintaining the differentiation of anti-inflammatory subgroups [101,102]. Naringenin also modulates cellular inflammatory responses by regulating the transcription of inflammatory factors and enhancing the lysosomal degradation of cellular inflammatory factors [103,104].

### 2.3. Microbiota Dysbiosis

SARS-CoV-2 leads to long-term changes in the microbiota, with decreased gut microbial diversity, fewer beneficial commensal bacteria, and increased opportunistic pathogens in patients [105]. Viral invasion dysregulates gut ecology by activating immunity, deregulating ACE2 expression, disrupting the intestinal barrier, and directly infecting bacteria [106,107]. Moreover, the antibiotics administered during treatment can cause more harm to the patient’s intestinal flora [108]. Furthermore, COVID-19 patients exhibited noteworthy alterations in the lung [109], oral [110], and nasopharyngeal microbiomes [111]. Changes in the microbiome persist for an extended period, and patients do not return to normal microbial abundance after 6 months of rehabilitation therapy [112]. Gut microbes remained significantly different from healthy controls after 3 months of recovery and showed a correlation with persistent symptoms of long COVID [113]. Changes in the composition and abundance of a patient’s gut microbes also affect their susceptibility to long COVID and complication severity [25]. These changes can have a serious impact on the health of the survivor, causing an inflammatory immune response in the body, changes in the levels of basal metabolite levels (e.g., amino acids, carbohydrates, and neurotransmitters), and clinical gastrointestinal symptoms [114]. Gut microflora dysbiosis can have neurological and pulmonary effects through cytokines and metabolites [115], which are mediated through the gut–lung and brain–gut axes [116,117].

Naringin and naringenin have been found to have beneficial regulatory effects on the composition and metabolism of gut flora [118,119]. The dietary intake of naringin and naringenin helps to control the intestinal microenvironment [120]. Naringin and naringenin can directly regulate the intestinal microbiota and its metabolism. Naringin significantly reduces the abundance of gastrointestinal disease-related bacteria (e.g., *Lachnoclostridium* and *Bilophila*) and enhances probiotic content in experiments [121]. Naringenin can regulate the growth and gene expression pattern of intestinal commensal microorganisms through interaction and activate genes related to cellular metabolism [122]. In addition, gut barrier dysfunction caused by microbiota dysbiosis may increase bacterial ectopia and contribute to host immune destabilization [123]. Naringin has been shown to reduce inflammation-associated protein expression and colonic damage caused by dextran sulfate sodium, as well as improve colonic barrier dysfunction [124].

### 2.4. Myalgic Encephalomyelitis/Chronic Fatigue Syndrome (ME/CFS)

ME/CFS is a chronic multisystem disease that often follows infectious diseases and may be associated with chronic viral infections such as human herpesviruses, human parvovirus B19V, and enteroviruses [125]. Patients with long COVID often experience persistent fatigue, myalgia, post-exertional malaise, insomnia, and exercise intolerance, with symptoms resembling the diagnostic criteria for ME/CFS [126]. Although there are no studies directly proving that acute COVID-19 triggers ME/CFS, several studies have identified multiple overlapping or similar symptoms between long COVID and ME/CFS [127]. COVID-19 also greatly increases the number of patients with ME/CFS [128], and bioinformatics analysis has identified a common network of genetic interactions between the two [129]. According to the diagnostic criteria for ME/CFS, the overall prevalence of ME/CFS was as high as 43% among patients with long COVID, with fatigue, malaise on exertion, and insomnia representing the main symptoms [130]. The invasion of SARS-CoV-2 significantly impacts both mitochondrial function and reserves, and the SARS-CoV-2 membrane protein causes mitochondrial apoptosis in epithelial cells [131]. Virus-induced decreases in mitochondrial membrane potential [132] and the downregulation of nuclear-encoded mitochondrial genes [133] have been observed in patients. In addition, the infiltration of amyloid-containing deposits and skeletal muscle injuries have also been observed in patients with long COVID [134]. The pathogenesis of ME/CFS involves neuroinflammation, redox imbalance, mitochondrial dysfunction, autoantibodies, and autonomic dysfunction [135,136,137].

The treatment of ME/CFS is still unclear and there are no effective treatments. Most treatments are based on the clinical condition and include central nervous system drugs, antiviral drugs, immunomodulators, analgesics, and nutritional supplements [138]. Fatigue and post-exercise discomfort may be due to reduced energy sources, mitochondrial dysfunction, and redox imbalance. Serum matrix metalloproteinase 9 (MMP-9) and muscle damage-associated lactate dehydrogenase (LDH) are associated with muscle fatigue. By stabilizing redox and elevating blood glucose levels, naringin effectively diminishes MMP-9 and LDH concentrations, thereby augmenting energy sources. This prolongs the duration of fatigue-inducing exercise and relieves post-exercise discomfort [139,140]. Moreover, naringenin can exert anti-fatigue effects by participating in the promotion of testosterone secretion [141]. Additionally, naringin can help to stabilize mitochondrial membrane potential, preserve mitochondrial integrity, and sustain mitochondrial complex activity [142,143], all of which can help to reduce mitochondrial dysfunction. Naringenin also protects neuronal mitochondrial function by activating the transcription factor Nrf2 [144]. In addition, naringin and naringenin are important redox regulators. Naringin significantly attenuated antigen-induced oxidative stress and reduced TNF-α levels in fatigued mice [145] while reinstating oxidative stress markers in rodent brains [146].

### 2.5. Myocarditis

Certain individuals who have survived the initial stages of COVID-19 experience enduring cardiac impairment following their recovery, ultimately resulting in myocarditis [147]. The clinical signs of myocarditis are heterogeneous and non-specific, typically consisting of chest pain, arrhythmias, generalized fatigue, dyspnea, and tachycardia [148]. The analysis of cardiac symptoms among COVID-19 survivors revealed that 9.79 percent displayed indications of chest pain, whereas 8.22 percent presented with arrhythmias [149]. Within a span of 2–3 months following SARS-CoV-2 infection, 19% of patients developed persistent symptoms of myocarditis [150]. One year after rehabilitation, patients had a significant 4.16-fold increased risk of myocarditis [151] and poorer clinical outcomes, with a mortality rate of 1.36% to 5% [152]. The direct pathogenic mechanism of myocarditis in long COVID is the direct interaction of SARS-CoV-2 with ACE2 receptors in cardiomyocytes and pericytes, causing immune dysregulation. Subsequently, immune dysregulation leads to cardiomyocyte injury and the release of inflammatory factors, such as IL-2, IL-6, IL-7, TNF-a, IFN-α/β, C-X-C motif cytokine 10 (CXCL10), and C-C motif ligand 2 (CCL2), which, in turn, induces persistent low-level inflammation [153]. Indirect pathogenic mechanisms include hypoxemia caused by the cytokine storm [154], high levels of antiheart autoantibodies caused by CD4^+^ T-lymphocytes [155], and cardiac mitochondrial dysfunction [156]. Myocardial injury can be identified by the presence of markers such as myoglobin, troponin, creatine kinase-MB, IL-6, LDH, and N-terminal pro-b-type natriuretic peptide [157].

Naringin and naringenin have been shown to have cardioprotective properties [45]. First, the anti-inflammatory properties of naringin can drastically reduce the inflammatory factors associated with cardiovascular injury, such as NF-κB, IL-6, IL-1β, and TNF-α, and suppress the inflammatory response [158]. Furthermore, the antioxidant effects of naringin can scavenge free radicals and increase the activity of antioxidant enzymes (superoxide dismutase and catalase) and GSH levels, effectively protecting the heart mitochondria from damage, thereby reducing DOX-induced apoptosis and vacuolization in cardiomyocytes [67]. In addition, naringin has the potential to alter the cellular channel currents in mouse ventricular myocytes, consequently exerting antiarrhythmic effects [159].

### 2.6. Pulmonary Fibrosis

Pulmonary fibrosis is one of the serious sequelae of COVID-19. It is estimated that 19% of patients recovering from COVID-19 have residual lung abnormalities [160]. The lungs are the primary target organ for viral invasion. SARS-CoV-2 causes severe lung damage, ultimately leading to post-COVID-19 pulmonary fibrosis (PCPF). Radiological manifestations of pulmonary fibrosis usually include bilateral lung infiltrates, “ground glass” opacity, and “honeycomb” lungs [161]. Its cellular and molecular characteristics include a decrease in lymphocytes and an increase in CRP and IL-6 [162]. Multiple studies have shown that the prevalence of PCPF among COVID-19 survivors is more than 9.3% [163]. Alveolar injury caused by SARS-CoV-2 leads to the secretion of pro-fibrotic and pro-inflammatory cytokines by alveolar macrophages and type 2 alveolar epithelial cells (AEC), including IFN-γ, transforming growth factor (TGF)-β, and IL-6 and IL-17. Cytokines induce myofibroblast differentiation by activating the WNT/β-catenin and YAP/TAZ pathways, ultimately leading to the combined effects of pulmonary fibrosis [164,165].

Naringin and naringenin are effective inhibitors of pulmonary fibrosis. The main target of current anti-pulmonary fibrosis therapy is the TGF-β/Smad pathway [166]. It has been shown that naringin inhibits cellular fibrosis by suppressing TGF-β overexpression and reducing downstream regulatory factor phosphorylation [167]. Moreover, the anti-inflammatory and antioxidant effects of naringin and naringenin can be used against pulmonary fibrosis [168]. Naringin has been shown to significantly reduce the infiltration of inflammatory cells induced by lipopolysaccharide and decrease the production of macrophage nitrogen monoxide (NO) and IL-6 [169]. In critically ill COVID-19 patients, naringenin also showed excellent IL-6 inhibition compared to synthetic monoclonal antibodies [46]. In a paraquat-induced acute lung injury model, naringin not only decreased the production of the inflammatory cytokines TNF-α and TNF-β1 but also inhibited oxidative stress by activating the expression of antioxidant enzymes (superoxide dismutase, glutathione peroxidase, and heme oxygenase 1) and regulated collagen formation by modulating the ratio of tissue inhibitors of metalloproteinases-1 (TIMP-1) to MMP-9, thereby preventing lung fibrosis [170]. Furthermore, naringin modulates the activating transcription factor 3/PTEN-induced kinase 1 (ATF3/PINK1) pathway and enhances mitophagy in lung tissues, resulting in the alleviation of bleomycin-induced idiopathic pulmonary fibrosis [171]. Naringenin also protects against cigarette-induced lung injury by regulating miRNAs in extracellular vesicles [172].

### 2.7. Cough

Coughing is the most common symptom of COVID-19 sequelae and is present in approximately 23% to 57% of COVID-19 survivors in all countries, and is highly prevalent in both mild and severe cases [173,174]. Cough can persist for months, with a prevalence of up to 2.5% even after one year of recovery [175], making it a major source of distress for patients. SARS-CoV-2 may induce chronic coughing through neuroinflammatory and neuroimmune mechanisms. Viral invasion causes the release of neuroinflammatory mediators (IL-1β, TNF, IFN, adenosine triphosphate (ATP)) and, in turn, activates vagal sensory neurons via transient receptor potential (TRP) channels. Sensory neurons release a variety of neuropeptides (calcitonin gene-related peptide, substance P, and neurokinin A), eventually leading to hypersensitivity of the cough pathway [176]. In addition, the underlying mechanisms of chronic cough may also be related to post-infectious lung abnormalities, cough underlying disease, upper airway cough syndrome, or cough-variant asthma (CVA) [177,178,179].

The current treatment of chronic coughing focuses on suppressing the cough reflex, reducing airway inflammation, drying and relieving coughing, and resolving phlegm to relieve coughing [180,181]. Naringin and naringenin have long been shown to have positive antitussive effects [182]. Naringenin can reduce inflammation caused by cigarette smoke in mice with chronic obstructive pulmonary disease by inhibiting the production of the pro-inflammatory factors IL-8, TNF-α, and MMP9, as well as the NF-κB pathway [183]. In an experimental CVA model, naringin has been shown to reduce irritation-induced cough and suppress the growth of airway-inflammatory factors (IL-4, IL-5, and IL-13) and leukocytes [184]. Moreover, naringin has been shown to significantly attenuate cigarette smoke-induced airway neurogenic inflammation by reducing substance P and NK-1 receptors [185]. Furthermore, naringin can be used in asthma treatment by promoting the proliferation of AECs through the bitter taste receptor (TAS2R)-related signal pathway, thereby inducing the relaxation of airway smooth muscle cells [186].

### 2.8. Diabetes

The occurrence of new-onset diabetes mellitus can be attributed to acute infection with SARS-CoV-2, and complications such as diabetic ketoacidosis (DKA) and hyperglycemic hyperosmolar syndrome are highly prevalent [187,188]. COVID-19 survivors have a 64–66% higher risk of developing diabetes than uninfected individuals [189]. The risk of developing diabetes in the first three months after COVID-19 infection was up to 95%, with a higher risk of developing type 2 diabetes (T2D) than type 1 diabetes (T1D) (70% and 48%, respectively) [190]. The data indicated that during the initial year of the COVID-19 outbreak, there was a notable surge of 9.5% and 25% in the occurrence of new pediatric T1Ds and DKAs, correspondingly, alongside a substantial rise in blood glucose levels [191]. A follow-up study found that most patients with new-onset T2D were cured 3 months after discharge from the hospital. In contrast, the remaining 37% of patients with T2D were diagnosed with persistent diabetes mellitus [192]. COVID-19-related abnormalities in glucose metabolism also recovered after one year [193]. On this bases, diabetes was also associated with serious illness, hospitalization, and death in COVID-19, so a bidirectional causal relationship between COVID-19 and diabetes was hypothesized [194]. Both COVID-19 patients and survivors exhibit insulin resistance and beta-cell dysfunction, which endure even after recuperation from the illness [195].

The pathogenesis of new-onset diabetes is mainly related to pancreatic autoimmunity, pancreatic injury, pro-inflammatory cytokine storms, the triggering of steroid drugs, and underlying diabetic activation of the body. T1D is an autoimmune disease, and the invasion of SARS-CoV-2 leads to autoimmune hyperactivation in the body [196]. Additionally, researchers have found pancreatic damage in some COVID-19 patients [197]. Thus there has been a hypothesis suggesting that ACE2 exhibits elevated expression levels in exocrine glands and pancreas islets [198] when SARS-CoV-2 enters them to induce pancreatic β-cell dysfunction or apoptosis, hindering insulin signaling [199]. T2D is mainly caused by short-term treatment and inflammation. Steroids, a widely prescribed medication for COVID-19, can potentially impact insulin sensitivity; nevertheless, the discontinuation of steroid therapy may lead to the recurrence of new-onset diabetes [200]. The inflammatory factor TNF has also been recognized as a common pathogenic target of COVID-19 and T2D [201].

Naringin and naringenin may exert antidiabetic potential by attenuating pancreatic β-cell damage or activating their proliferation. Naringenin protects pancreatic β-cells from streptozotocin (STZ)-induced immune stress by activating Nrf2, leading to a substantial decrease in blood glucose levels and restoring normal insulin levels [202]. Naringin inhibits mitochondria-mediated and death receptor-mediated apoptosis in pancreatic β cells [203]. Forkhead box M1 (FoxM1) transcription factor affects pancreatic adult beta cell proliferation, and naringin increases beta cell mass and treats diabetes by upregulating FoxM1 [204]. Naringin and naringenin also showed significant ameliorative effects on insulin resistance. In the an STZ- and a nicotinamide-induced T2D model, naringin and naringenin enhanced insulin secretory responses and the expression of insulin receptors and their sensitizers [205]. In T2D patients, naringin was found to improve glucose metabolism, increase residual insulin secretion, and stimulate glycogen synthesis; however, absolute insulin deficiency prevented it from regulating glucose levels in T1D patients, yet it could reverse T1D-induced DKA [206]. Furthermore, naringin and naringenin regulate carbohydrate and lipid metabolism. Naringin modulates the activity of glycolysis-related enzymes and regulates intestinal carbohydrate absorption [207]. Naringenin can reduce metabolic disorders by modulating immune-related inflammatory factors and inhibiting the infiltration of inflammatory cells into adipose tissue [208]. In addition, naringenin has a mitigating effect on the complications of diabetes. Naringenin, when used in conjunction with insulin, modulates matrix metalloproteinases to reduce neuropathic pain caused by diabetes [209]. Naringenin inhibits high-glucose-induced vasculopathy by downregulating the hyperproliferation and migration of vascular smooth muscle cells [210].

### 2.9. Pain

COVID-19 patients often have accompanying pain, including headache, musculoskeletal pain, and testicular pain, during the acute infection period and after rehabilitation [211,212]. Although the majority of pain symptoms subside two months after recovery, 10% of COVID-19 survivors still experience persistent musculoskeletal muscle pain, and the length of the pain does not correlate with the intensity of COVID-19 [213]. The post-infection follow-up of COVID-19 survivors unveiled a range of symptoms, encompassing general pain (13.40%), joint pain (28.25%), muscle pain (13.30%), headache (9.50%), and chest pain (12.12%) [214]. Pain development is mainly associated with direct action through ACE2 receptors [215], the inflammatory cytokine storm [216], and the driving effect of prostaglandins [217]. ACE2 receptors are abundant in skeletal muscle and other organs, resulting in viral harm to the muscles, and their neurophilicity can also lead to neuronal damage, resulting in pain [218]. The virus triggers an extended inflammatory reaction, resulting in heightened hyperexcitability of the central nervous system, thereby exacerbating pain [219]. COVID-19 patients experiencing headache symptoms exhibited elevated concentrations of inflammatory molecules and harmful molecules (high-mobility group protein B1 (HMGB1), NOD-like receptor thermal protein domain-associated protein 3 (NLRP3), and IL-6), potentially contributing to the initiation of trigeminal activation [220,221].

Naringin and naringenin have effective analgesic effects. Naringin and naringenin can attenuate neuronal stimulation directly by inhibiting the inflammatory response. Naringin attenuates iodoacetate-induced osteoarthritis pain by inhibiting the secretion of adrenaline and pro-inflammatory factors (IL-6, NO, and TNF-α) [222]. The administration of naringenin in a rat model of neuropathic pain resulted in the inhibition of glial cell activation caused by spinal nerve ligation injury, as well as a reduction in elevated levels of inflammatory factors (TNF-α, IL-1β, and monocyte chemotactic protein 1) [223]. Moreover, naringin and naringenin can directly modulate the damage perception pathway to achieve analgesic effects. Transient receptor potential vanilloid member 1 (TRPV1) is associated with noxious temperature sensation and inflammatory pain, and its antagonists have been considered potential analgesics [224,225,226]. By interacting with TRPV1, naringin suppresses nervous system hyperexcitability and potentially mitigates nerve pain by inhibiting oxidative stress [227]. Transient receptor potential melastatin-3 (TRPM3) ion channels are involved in organismal injury perception and are expressed on somatosensory neurons. Naringenin can effectively block TRPM3 channels in vivo and in vitro, thus supporting its analgesic effect [228]. Moreover, naringenin inhibits superoxide anion-induced inflammatory pain by activating the NO-cGMP-PKG-KAP signaling pathway and reducing nociceptive cytokine expression [229].

### 2.10. Reproductive Dysfunction

The acute infection phase of COVID-19 leads to reproductive dysfunction in both genders, including erectile dysfunction, orchitis, reduced testosterone levels, ovarian dysfunction, and menstrual changes [230,231]. After 3.8 months of recovery, 35.9% of men and 27.7% of women were found to have sexual dysfunction, with different mechanisms in both cases [232]. A continued high prevalence of erectile dysfunction in men was observed 3 months after COVID-19 recovery [233]. The impact of long COVID on women’s reproductive health is mainly characterized by menstrual irregularities, gonadal dysfunction, and fertility problems [234]. Both ACE2 and transmembrane serine protease 2 (TMPRSS2) are highly expressed in the gonads, with higher expression in male gonads and detectable SARS-CoV-2 infection in the testis [235]. Therefore, it is hypothesized that the gonads, especially the testes, are susceptible to SARS-CoV-2 infection and damage. Male long COVID patients with coagulation abnormalities and endothelial damage also experience testicular inflammation [236]. Testicular injury, oxidative stress by reactive oxygen species [237], and decreased testosterone due to dysfunction of the hypothalamic–pituitary–gonadal axis [238] can lead to erectile dysfunction [239]. Moreover, the autoimmune inflammatory response of the body can also cause a decrease in sperm quality [240].

Naringenin also has an attenuating effect on testicular and sperm damage. Naringenin can reduce the harmful effects of bisphenol A on the testes by suppressing oxidative stress and mitochondrial apoptosis [241]. Naringenin has been shown to reduce the harmful effects of antiretroviral medication on the male reproductive system of rats while preserving the normal physical structure of the testes and sperm viability [242]. Furthermore, naringenin has a protective effect on the ovary and can alleviate polycystic ovary syndrome. In vivo naringin treatment in rats inhibits steroidogenic enzymes and consequently controls ovulation, thereby restoring ovarian morphology and cystic follicle levels in patients with polycystic ovary syndrome [243].

### 2.11. Thrombus Formation

Endothelial damage caused by SARS-CoV-2 can lead to coagulation dysfunction in certain patients, ultimately resulting in the development of long-term concomitant thrombosis [244]. Individuals afflicted with long COVID face an elevated likelihood of experiencing arterial thromboembolic events and venous thromboembolic events. They are susceptible to acute pulmonary embolism (PE), deep vein thrombosis (DVT), myocardial infarction, and acute respiratory distress syndrome (ARDS) [245,246]. COVID-19 survivors had a significantly higher incidence of thrombophilia, PE (2.5–6.3%), and DVT (1.2–6.4%), which was 2 to 3 times higher than in uninfected individuals [247]. Long COVID-induced thrombosis can be caused by various factors, such as damage to the endothelium, the abnormal production of fibrin during platelet aggregation, and dysregulation of the immune system [248]. SARS-CoV-2 invasion can promote endothelial damage and dysfunction through direct interaction with ACE2 receptors [244]. Its spiny proteins can also interact with platelets and fibrin, inducing the formation of fibrin-like microclots, which are difficult to hydrolyze [249]. Moreover, viral invasion inducing a storm of inflammatory factors (IL-1β, IL-6, and TNF) activates exogenous coagulation pathways and promotes the inhibition of anticoagulant routes [250]. In addition, the persistence of the virus in the outer vesicles, the formation of autoantibodies, and chronic hypoxia can lead to long-term coagulation disorders [251]. Patients with long COVID who displayed coagulation abnormalities exhibited increased levels of hepatocyte growth factor (HGF), IL-6, and D-dimer [252].

Naringin can exert its antithrombotic effect by protecting endothelial cells, inhibiting platelet activation, and inhibiting thrombin activity. Naringin has been shown to reduce inflammation and alter the permeability of endothelial cells, thus mitigating endothelial dysfunction [169]. Naringin protects endothelial cell function by upregulating NO bioavailability [253], decreasing levels of inflammatory factors (IL-1β, IL-6, and IL-18), and reversing YAP downregulation [254]. Naringin inhibits excessive autophagy in endothelial cells by activating the PI3K-Akt-mTOR pathway [255]. Moreover, naringenin can impede platelet aggregation. Naringenin has a structure-dependent inhibitory effect on platelet function in both whole blood and plasma [256], and can also inhibit platelet activation by targeting the PI3K/Akt pathway, preventing FeCl_3_-induced carotid artery thrombosis and vascular occlusion and displaying effective antithrombotic properties [257]. In addition, research on molecular docking demonstrated that naringenin is strongly attracted to thrombin and can attach itself to the active core of thrombin, potentially impeding its activity [258]. Protein disulfide isomerase (PDI) in plasma is involved in the conformational formation of coagulation-associated proteins, and naringin can also affect thrombus formation and stabilization by binding to PDI and inducing conformational changes [259].

## 3. Conclusions and Prospects

This review addresses the pathogenesis and statistical status of eleven complications of long COVID and specifically addresses the therapeutic potential of naringin and naringenin in long COVID (Table 1). The clinical symptoms and biomarkers of long COVID have been widely reported, and long COVID can indeed have serious long-term effects on human health [260,261]. There are no definitive medications that directly treat multiple long COVID complications [262], and the treatment approach for long COVID is still based solely on clinical symptoms and the use of nutraceuticals and probiotics to improve symptoms [263]. Therefore, treating patients with multiple syndromes requires a combination of drugs. Naringin and naringenin, as important constituents of medicinal plants, are characterized by strong safety, multiple targets, and few side effects, with excellent performance in treating various diseases. They demonstrate protection against biotic stress by inducing a hormonal dose response in various cell models [264]. Moreover, it is worth noting that nano-preparations of natural products such as naringenin can improve their bioavailability and drug targeting after oral administration [265]. We enumerate the therapeutic and palliative effects of naringin and naringenin on conditions with symptoms or pathogenic mechanisms similar to those of long COVID, which are based on their anti-inflammatory, antimicrobial, antiviral, anti-free radical, cardiovascular-protection, microbiota-modulation, and neuron-protection effects. The purpose of this review is to provide a theoretical foundation for the use of naringin and naringenin as potential treatments for long COVID. Further research is necessary to ascertain the true efficacy and appropriate dosage of naringin and naringenin in treating long COVID, as well as conduct thorough follow-up clinical trials.

## Figures and Tables

**Figure 1 microorganisms-12-00332-f001:**
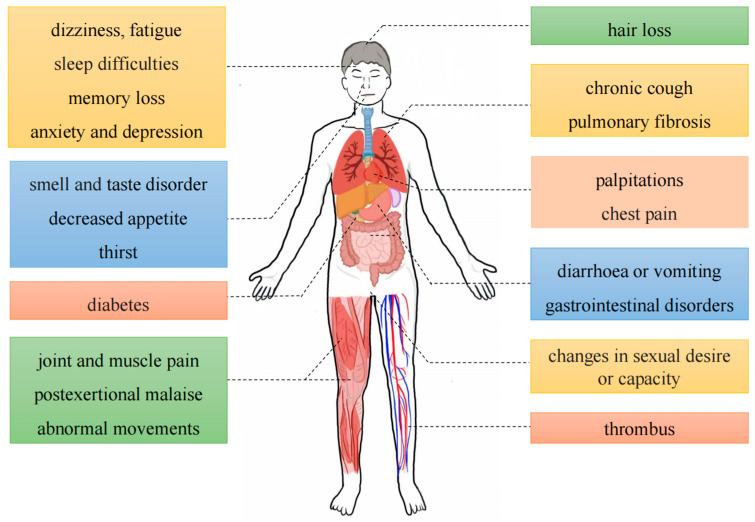
Long COVID symptoms.

**Figure 2 microorganisms-12-00332-f002:**
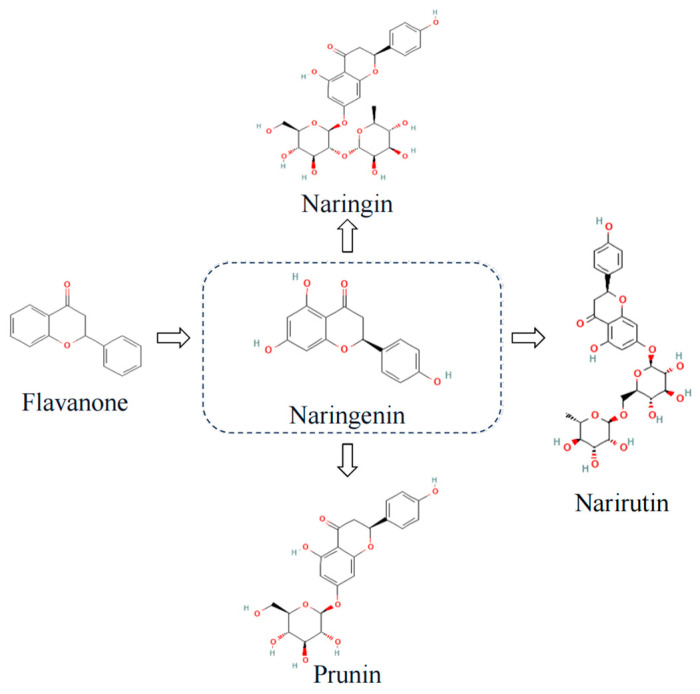
Chemical structures of flavanone, naringenin, and its derivatives. The 2D structure images were obtained from PubChem (https://pubchem.ncbi.nlm.nih.gov).

**Figure 3 microorganisms-12-00332-f003:**
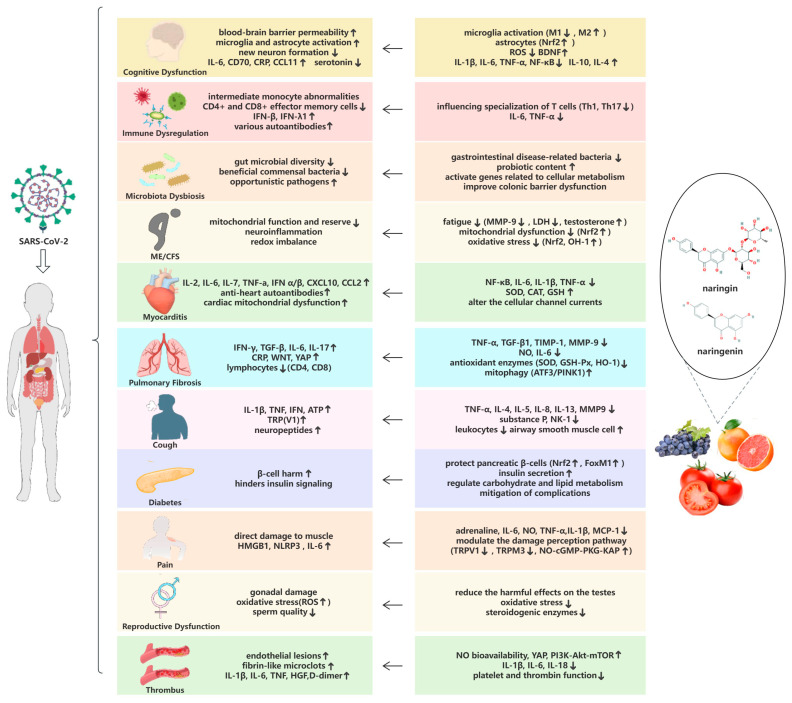
Summary of long COVID symptom pathogenesis (**left**) and potential therapeutic mechanisms (**right**). Upward arrows (↑) in the colored panels of the table indicate increases and downward arrows (↓) indicate decreases. Leftward arrows (←) between the panels indicate the palliative and therapeutic effects of naringin on long COVID symptoms.

**Table 1 microorganisms-12-00332-t001:** Potential pathophysiological mechanisms of long COVID symptoms and the functions of naringin and/or naringenin.

Long COVID Symptoms	Potential Pathophysiological Mechanisms of Long COVID	Functions of Naringin and/or Naringenin
Cognitive Dysfunction	Interference with blood–brain barrier led to neuronal damage [56,57] Downregulation of new neuron formation [59] Pro-inflammatory factors led to depression [60] Direct damage to the olfactory epithelium [63]	Modulation of glial cell activation [69,71,72] Enhanced synaptic plasticity [73] Protected the nervous system from oxidative stress and inflammation [77] Inhibited hippocampal apoptosis [80] Enhanced the endocrine nervous system [81]
Immune Dysregulation	Intermediated monocyte abnormalities and T-cell activation [87,88] Decline in CD4^+^ and CD8^+^ effector memory cells [22] Lack of dendritic cells [90] Reduction in non-classical monocyte and lymphocyte subsets [91] Increased levels of autoantibodies [93]	Influenced growth and specialization of T cells [98,99,100,101,102] Regulated the transcription and lysosomal degradation of inflammatory factors [103,104]
Microbiota Dysbiosis	Changes in the composition and abundance of the gut [105], lung [109], oral [110], and nasopharyngeal microbiomes [111] Altered basal metabolite levels [114]	Controlled the intestinal microenvironment [120] and colonic barrier [124]. Regulated the intestinal microbiota [121] Regulated gene expression pattern of intestinal commensal microorganisms [122]
ME/CFS	Impacted both mitochondrial function and reserves [131,132,133] Infiltration of amyloid-containing deposits [134]	Stabilization of energy sources [139,140,141] Reduced mitochondrial dysfunction [142,143,144] Attenuated oxidative stress [145,146]
Myocarditis	Inflammation of cardiomyocytes through direct action of SARS-CoV-2 [153] Hypoxemia [154] High levels of antiheart autoantibodies [155] Cardiac mitochondrial dysfunction [156]	Reduced the inflammatory factors associated with cardiovascular injury [158] Protected cardiac mitochondria from oxidative damage [67] Altered the cellular channel currents [159]
Pulmonary Fibrosis	Alveolar injury led to secretion of pro-fibrotic and pro-inflammatory cytokines [164,165]	Inhibition of TGF-β overexpression [167] Reduced the infiltration of inflammatory cells [169] Regulated collagen formation [170] Enhanced mitophagy in lung tissues [171] Regulated miRNAs in extracellular vesicles [172]
Cough	Neuroinflammatory mediators led to hypersensitivity of the cough pathway [176]	Inhibited the production of pro-inflammatory factors [183,184] Attenuated airway neurogenic inflammation [185] Induced relaxation of airway smooth muscle cells [186]
Diabetes	Autoimmune hyperactivation in the body [196] β-cell harm and pancreatic damage [197,199] Steroids impacted insulin sensitivity [200]	Attenuated pancreatic β-cell damage or activated their proliferation [202,203,204] Ameliorative effect on insulin resistance [205,206] Regulated carbohydrate and lipid metabolism [207,208] Mitigated effect on the complications of diabetes [209,210]
Pain	Direct damage to muscle by the virus [218] Inflammation led to overstimulation of the nervous system [219,220,221]	Attenuated neuronal stimulation by inhibiting the inflammatory response [222,223] Modulated the damage perception pathway [227,228,229]
Reproductive Dysfunction	Virus-induced gonadal damage [235] Coagulation abnormalities and endothelial damage [236] Decreased testosterone [238] Decrease in sperm quality caused by inflammatory response [240]	Inhibition of oxidative stress and mitochondrial apoptosis in testes [241] Preserved the structure of the testes and sperm viability [242] Inhibited steroidogenic enzymes and controlled ovulation [243]
Thrombus Formation	Endothelial damage and dysfunction [244] Formation of fibrin-like microclots that are difficult to hydrolyze [249] Inflammatory factor storm activated exogenous coagulation pathways [250] Formation of autoantibodies and chronic hypoxia [251]	Upregulated NO bioavailability [253] Decreasing levels of inflammatory factors [254] Inhibited excessive autophagy in endothelial cells [255] Inhibited platelet activation [256,257] Inhibited thrombin activity [258,259]

## Data Availability

Not applicable.

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
