# Peer review of "Potential Beneficial Effects of Naringin and Naringenin on Long COVID—A Review of the Literature"

_microorganisms, 2024, doi:10.3390/microorganisms12020332_

Round 1

Reviewer 1 Report

Comments and Suggestions for Authors

The manuscript is an interesting review, dealing with the potential beneficial effects of naringin and naringin in treating long COVID symptoms and conditions considering the activity previously shown by the compounds in treating several related disorders.

The manuscript fits the purpose of the journal and the data presented could sound interesting  to the scientific community. However, it needs to be improved before it can be considered for publication. Here are some suggestions:

-            Writing should be improved, and language should be more appropriate and fluent by minimizing term repetition. I suggest having the entire text carefully proofread by a native English speaker.

-            Add a brief introduction to paragraph 2 and move the figure here (as an introduction to the subsequent paragraphs).

-            Rearrange the order of the disorders examined (for instance, starting with those related to the CNS and then moving on to the peripheral ones).

-            References need to be rearranged:

-            ​all references must be rechecked, for example Ref. 34, 72, 279 and so on.

-            some missing references are required and need to be introduced while unnecessary ones removed. In detail:

-             add after ref 37:

“Sub-micromolar inhibition of sars-cov-2 3clpro by natural compounds” Pharmaceuticals, 2021, 14(9), 892;

and, after ref 43:

-             

“A pilot study on the nutraceutical properties of the Citrus hybrid Tacle® as a dietary source of polyphenols for supplementation in metabolic disorders” Journal of Functional Foods, 2019, 52, pp. 370–381.

-            Limit, as much as possible, references that do not refer to naringin, naringenin, or COVID.
For example, in citations 66 and 67, exclude ref. 67, and so on.

After these modification the paper can be accepted for publication.

Comments on the Quality of English Language

The manuscript is an interesting review, dealing with the potential beneficial effects of naringin and naringin in treating long COVID symptoms and conditions considering the activity previously shown by the compounds in treating several related disorders.

The manuscript fits the purpose of the journal and the data presented could sound interesting  to the scientific community. However, it needs to be improved before it can be considered for publication. Here are some suggestions:

-            Writing should be improved, and language should be more appropriate and fluent by minimizing term repetition. I suggest having the entire text carefully proofread by a native English speaker.

-            Add a brief introduction to paragraph 2 and move the figure here (as an introduction to the subsequent paragraphs).

-            Rearrange the order of the disorders examined (for instance, starting with those related to the CNS and then moving on to the peripheral ones).

-            References need to be rearranged:

-            ​all references must be rechecked, for example Ref. 34, 72, 279 and so on.

-            some missing references are required and need to be introduced while unnecessary ones removed. In detail:

-             add after ref 37:

“Sub-micromolar inhibition of sars-cov-2 3clpro by natural compounds” Pharmaceuticals, 2021, 14(9), 892;

and, after ref 43:

-             

“A pilot study on the nutraceutical properties of the Citrus hybrid Tacle® as a dietary source of polyphenols for supplementation in metabolic disorders” Journal of Functional Foods, 2019, 52, pp. 370–381.

-            Limit, as much as possible, references that do not refer to naringin, naringenin, or COVID.
For example, in citations 66 and 67, exclude ref. 67, and so on.

After these modification the paper can be accepted for publication.

Author Response

Response to Reviewer 1 Comments

 Dear Reviewer,

On behalf of my co-authors, we thank you very much for giving us valuable comments on our manuscript entitled “Potential beneficial effects of naringin and naringenin on long COVID-A review of the literature (microorganisms-2841110)”. Those comments are all valuable and very helpful for revising and improving our manuscript, as well as the important guiding significance to our future work. We have tried our best to revise our manuscript according to your comments and make our responses point by point. Here are our responses.

Comments 1: Writing should be improved, and language should be more appropriate and fluent by minimizing term repetition. I suggest having the entire text carefully proofread by a native English speaker.

Response 1: We have tried our best to improve the English language in the revised manuscript. Meanwhile, we work on writing improvement with advice from MDPI's English editing.

Comments 2: Add a brief introduction to paragraph 2 and move the figure here (as an introduction to the subsequent paragraphs).

Response 2: Thanks for your suggestion. We have moved the location of the figure (Figure 3) to the beginning of the second paragraph, as suggested, and added the content as follows:

“In the following, we will discuss the pathogenesis of 11 common symptoms in patients with long COVID and make reasonable assumptions about the potential therapeutic role of naringin and naringenin in treating these symptoms (Figure 3).” (page 4, paragraph 1, and line 113)

Comments 3: Rearrange the order of the disorders examined (for instance, starting with those related to the CNS and then moving on to the peripheral ones).

Response 3: We strongly agree with your suggestion and have changed it to the following order: cognitive dysfunction, immune dysregulation, microbiota dysbiosis, myalgic encephalomyelitis/ chronic fatigue syndrome, myocarditis, pulmonary fibrosis, cough, diabetes, pain, reproductive dysfunction, thrombus.

Comments 4:  References need to be rearranged. All references must be rechecked, for example Ref. 34, 72, 279 and so on.

Some missing references are required and need to be introduced while unnecessary ones removed. In detail:  add after ref 37: “Sub-micromolar inhibition of sars-cov-2 3clpro by natural compounds” Pharmaceuticals, 2021, 14(9), 892; and, after ref 43: “A pilot study on the nutraceutical properties of the Citrus hybrid Tacle® as a dietary source of polyphenols for supplementation in metabolic disorders” Journal of Functional Foods, 2019, 52, pp. 370–381.

Response 4: We gratefully appreciate your comment. We have checked all the references and corrected the formatting of the problematic references in them.

We have revised the original references 34, 72, and 279. As a result of the reorganization of paragraphs, the serial numbers now read 34, 182, and 86, respectively. Article 34 reference are preprints and have not been peer-reviewed. It has now been replaced. The references have been modified as follows:

  1. Imran, M.; Khan, S.A.; Abida, null; Alshammari, M.K.; Alkhaldi, S.M.; Alshammari, F.N.; Kamal, M.; Alam, O.; Asdaq, S.M.B.; Alzahrani, A.K.; et al. Nigella Sativa L. and COVID-19: A Glance at The Anti-COVID-19 Chemical Constituents, Clinical Trials, Inventions, and Patent Literature. Molecules 2022, 27, 2750, doi:10.3390/molecules27092750. (page 17, paragraph 3, and line 660)
  2. Zanasi A, Fontana GA, Mutolo D. Cough: Pathophysiology, Diagnosis and Treatment. 1st ed. Springer International Publishing 2020:181. ISBN 10: 3030485714. https://doi.org/10.1007/978-3-030-48571-9. (page 27, paragraph 7, and line 1086)
  3. Klein, J.; Wood, J.; Jaycox, J.R.; Dhodapkar, R.M.; Lu, P.; Gehlhausen, J.R.; Tabachnikova, A.; Greene, K.; Tabacof, L.; Malik, A.A.; et al. Distinguishing Features of Long COVID Identified through Immune Profiling. Nature 2023, 623, 139–148, doi:10.1038/s41586-023-06651-y. (page 20, paragraph 12, and line 810)

    We have added the missing references, which are necessary for the rigor of our review.

39. Rizzuti, B.; Ceballos-Laita, L.; Ortega-Alarcon, D.; Jimenez-Alesanco, A.; Vega, S.; Grande,F.; Conforti, F.; Abian, O.; Velazquez-Campoy, A. Sub-Micromolar Inhibition of SARS-CoV-2 3CLpro by Natural Compounds. Pharmaceuticals (Basel). 2021, 14, 892, doi:10.3390/ph14090892. (page 17, paragraph 8, and line 676)
45. Casacchia, T.; Occhiuzzi, M.A.; Grande, F.; Rizzuti, B.; Granieri, M.C.; Rocca, C.; Gattuso, A.; Garofalo, A.; Angelone, T.; Statti, G. A Pilot Study on the Nutraceutical Properties of the Citrus Hybrid Tacle® as a Dietary Source of Polyphenols for Supplementation in Metabolic Disorders. J. Funct. Foods 2019, 52, 370–381, doi:10.1016/j.jff.2018.11.030. (page 17, paragraph 14, and line 693)

At the same time, we have removed unnecessary references that are similar in citation content or could be covered by updated literature. A total of 37 articles have been removed.

Comments 5: Limit, as much as possible, references that do not refer to naringin, naringenin, or COVID. For example, in citations 66 and 67, exclude ref. 67, and so on.
Response 5: We appreciate your comment. We have tried our best to cut down the references without affecting the main content of the article. The original total number of references was 304, which has now been reduced to 266, a reduction of 38 references.

Reviewer 2 Report

Comments and Suggestions for Authors

Dear authors

The review here is focused on the characteristics and applications on naringenin against covid and others. Have have the following points to highlight as modifications of the text:

1. add some figures, at least of naringenin and consider to add also the natural compounds family belongng to.

2. add schemes and tables on biological data as reminder for the readers

3. the application of such natural compounds has been already reviewed and studied by others, thus I suggest to implement the recent findings on this aspect, for example: Peptide Human Neutrophil Elastase Inhibitors from Natural Sources: An Overview; Nanoformulations of natural products for management of metabolic syndrome".

Comments on the Quality of English Language

none

Author Response

Response to Reviewer 2 Comments

Dear Reviewer,

Thank you very much for giving us valuable comments on our manuscript entitled “Potential beneficial effects of naringin and naringenin on long COVID-A review of the literature (microorganisms-2841110)”. Those comments are all valuable and very helpful for revising and improving our manuscript, as well as the important guiding significance to our future work. We have tried our best to revise our manuscript according to your comments and make our responses point by point. Here are our responses.

Comments 1: Add some figures, at least of naringenin and consider to add also the natural compounds family belongng to.

Response 1: We agree with this comment. Therefore, we have added a figure (Figure 2) to the end of the fourth paragraph of the introduction as suggested (page 3, paragraph 2, and line 106). The figure shows the chemical structure of flavanone, naringenin, and its derivative. In addition, we have added a figure of the symptoms of long COVID (Figure 1) to make it easier for the reader to understand (page 2, paragraph 2, and line 72).

Comments 2: Add schemes and tables on biological data as reminder for the readers.

Response 2: We appreciate your comment. We have added a table with the potential pathophysiological mechanisms and functions of naringin and/or naringenin for eleven long COVID symptoms as suggested (page 12, paragraph 2, and line 528).

Comments 3: The application of such natural compounds has been already reviewed and studied by others, thus I suggest to implement the recent findings on this aspect, for example: Peptide Human Neutrophil Elastase Inhibitors from Natural Sources: An Overview; Nanoformulations of natural products for management of metabolic syndrome".

Response 3: We gratefully appreciate your comment. We likewise consider the recent findings on the application of natural compounds (especially naringenin and naringenin) to be essential for our review. Therefore, we have added the recent findings on this aspect. It is expressed in the text as:

“Especially, natural and semi-synthetic flavonoids and neutrophil elastase inhibitors isolated from natural sources show great advantages in fighting SARS-CoV-2 or treating long COVID [38–40].” (page 3, paragraph 1, and line 81)

“Moreover, it is worth noting that nano-preparations of natural products such as naringenin can improve their bioavailability and drug targeting after oral administration [266].” (page 14, paragraph 2, and line 543)

References:

40. Marinaccio, L.; Stefanucci, A.; Scioli, G.; Della Valle, A.; Zengin, G.; Cichelli, A.; Mollica, A. Peptide Human Neutrophil Elastase Inhibitors from Natural Sources: An Overview. Int. J. Mol. Sci. 2022, 23, 2924, doi:10.3390/ijms23062924.

266. Davatgaran Taghipour, Y.; Hajialyani, M.; Naseri, R.; Hesari, M.; Mohammadi, P.; Stefanucci, A.; Mollica, A.; Farzaei, M.H.; Abdollahi, M. Nanoformulations of Natural Products for Management of Metabolic Syndrome. Int. J. Nanomedicine 2019, 14, 5303–5321, doi:10.2147/IJN.S213831.

Round 2

Reviewer 1 Report

Comments and Suggestions for Authors

The manuscript has been improved and can now be accepted for publication.

Comments on the Quality of English Language

 Only, minor editing of the English language is required.